# Changes in Air Pollution-Related Behaviour Measured by Google Trends Search Volume Index in Response to Reported Air Quality in Poland

**DOI:** 10.3390/ijerph182111709

**Published:** 2021-11-08

**Authors:** Wojciech Nazar, Katarzyna Plata-Nazar

**Affiliations:** 1Faculty of Medicine, Medical University of Gdańsk, Marii Skłodowskiej Curie 3a, 80-210 Gdańsk, Poland; 2Department of Pediatrics, Gastroenterology, Allergology and Nutrition, Medical University of Gdańsk, Nowe Ogrody 1-6, 80-803 Gdańsk, Poland; knazar@gumed.edu.pl

**Keywords:** particulate matter, Poland, air pollution, social awareness, search volume index

## Abstract

Decreased air quality is connected to an increase in daily mortality rates. Thus, people’s behavioural response to sometimes elevated air pollution levels is vital. We aimed to analyse spatial and seasonal changes in air pollution-related information-seeking behaviour in response to nationwide reported air quality in Poland. Google Trends Search Volume Index data was used to investigate Poles’ interest in air pollution-related keywords. PM_10_ and PM_2.5_ concentrations measured across Poland between 2016 and 2019 as well as locations of monitoring stations were collected from the Chief Inspectorate of Environmental Protection databases. Pearson Product-Moment Correlation Coefficients were used to measure the strength of spatial and seasonal relationships between reported air pollution levels and the popularity of search queries. The highest PM_10_ and PM_2.5_ concentrations were observed in southern voivodeships and during the winter season. Similar trends were observed for Poles’ interest in air pollution-related keywords. Greater interest in air quality data in Poland strongly correlates with both higher regional and higher seasonal air pollution levels. It appears that Poles are socially aware of this issue and that their intensification of the information-seeking behaviour seems to indicate a relevant ad hoc response to variable threat severity levels.

## 1. Introduction

Air pollution is defined as the contamination of the environment by any chemical, physical or biological agent that modifies the natural characteristics of the atmosphere [1]. It can be divided into indoor and outdoor (ambient) air pollution. The most common indicators of air pollution are the atmospheric concentration of particles with a diameter of 10 microns or less and 2.5 microns or less, described respectively as coarse particulate matter 10 (PM_10_) and fine particulate matter 2.5 (PM_2.5_). These particles are composed of organic and elemental carbon, nitrates, sulphates and trace elements, including nickel, cadmium, arsenic and lead particles [2,3]. The World Health Organisation (WHO) sets the guideline values at 20 μg·m^−3^ annual mean and 50 μg·m^−3^ 24-h mean for PM_10_ concentration and 10 μg·m^−3^ annual mean and 25 μg·m^−3^ 24-h mean for PM_2.5_ concentration [4].

In 2016, about 90% of the world population was breathing polluted air [5]. The highest levels of air pollution are found in low-income and middle-income countries. In Europe, the value of the PM_10_ annual mean concentration set by the WHO guidelines was exceeded at 51% of the stations supervised by the European Environment Agency and in all the reporting countries, except Estonia, Finland and Ireland [5]. As 36 of the 50 most polluted cities in the European Union are in Poland [6], the issue of air pollution is of particular importance in this country. The greatest contributors to the problem are the coal-based economy and household heating systems [6]. Ambient air pollution is estimated to cause about 4,200,000 premature deaths worldwide [1,4].

It is of high importance that individuals exposed to polluted air take active steps to reduce associated risks. To reduce direct exposure to high outdoor particulate matter concentrations people can, for example, avoid exercising outdoors or close to high-traffic areas [7], stay indoors and close windows, or utilise air purifiers. Studies carried out in China and the US found that people reduced their outdoor activity when atmospheric PM_2.5_ concentration rose [8,9,10]. Moreover, an increase in PM_2.5_ concentration in China is associated with higher online searches for anti-PM_2.5_ masks and air filters [11]. Other research showed that individuals limited their transportation-related physical activity and spent more time at home when particulate matter concentrations increased [9,12]. These results show that people not only search for the information on their local air quality index but use it as a guideline to adapt their behavioural patterns. To date, no such study has been carried out in Poland [13].

Despite growing general awareness and specific information-seeking and self-protection behaviours, an increased number of hospital admissions correlates with the worse air quality in Poland [14,15]. A similar trend is observed worldwide [16,17,18]. Moreover, it was demonstrated that short-term exposure to polluted air correlates with increased mortality [19]. This includes cardiovascular and respiratory diseases’ cause-specific mortality rates [14,15,20,21].

However, before active prevention, individuals must understand what air pollution is in general. It may be possible to achieve this passively, for instance by governmental promotion of the issue [22]. This said, what would likely be more effective is active and purposeful information-seeking, self-education and acquisition of air pollution awareness by individuals. This process can be illustrated by, for example, societal emphasis on the importance of checking your local daily air quality index. This in turn can be measured by investigating the popularity of relevant search terms entered into internet search engines, for example with the use of the Google Trends Search Volume Index (GTSVI). It shows the level of interest in a given search term, by a specifically located population within the specified period. If these data are compared to the reported national air quality data, an analysis of people’s response to reported air pollution levels can be performed. 

The primary aim of this study is to analyse spatial and seasonal changes in air pollution-related information-seeking behaviour measured by the prevalence of specific keywords in the GTSVI, in response to nationwide reported air quality in Poland.

Secondary objectives include a description of the aerial distribution of air quality monitoring stations and a simple spatial and seasonal analysis of changes in air quality in Poland.

## 2. Materials and Methods

### 2.1. Keyword Search

First, the authors proposed keywords related to air pollution. As Google Trends displays only a relative level of interest in a given search term, keywords were entered into KWFinder [23], a keyword research tool, part of a suite of search engine optimization (SEO) tools developed by Mangools, to find words with significant general search volume based on absolute values.

After initial analysis, seven keywords were selected. Subsequently, keywords were entered into Google Trends [24], to generate data. The search was performed in Polish. Three of the initially selected keywords: “air purity” (*czystość powietrza*), “PM_2.5_” and “PM_10_” were rejected, as data related to these keywords was available only nationally, making voivodeship-level spatial analysis impossible. Eventually, four keywords were selected: “smog” (*smog*), “air pollution” (*zanieczyszczenie powietrza*), “air quality” (*jakość powietrza*) and “air purifier” (*oczyszczacz powietrza*). Based on the KWFinder data, the average annual search volumes for these keywords were 19,900, 11,000, 8600 and 66,700, respectively. This amounts to a total number of about 420,000 searches over four years. The validity of the selection as related to air pollution was confirmed by checking that entering them into the Google search engine [25] resulted in a display of general information about smog and air pollution (1st keyword), particulate matter concentration maps and pollution alerts (2nd and 3rd keyword) or air purifiers (4th keyword).

Keyword-specific daily search Index between 1 January 2016 and 31 December 2019 was extracted from Google Trends for further analyses. This period was selected as at the beginning of 2016, when Google introduced an improvement to their Google Trends data collection system [24]. The year 2020 was not included as the ongoing COVID-19 pandemic [26] may have affected people’s behaviour which would result in biased data. 

Google Trends shows interest in a given keyword using a search volume index (SVI). The relative value indicating the popularity of a term on a selected day ranges between 0 and 100. A value of 100 is ascribed to the day during which a keyword was searched the greatest number of times in the given investigated time frame (in this case between 2016 and 2019). A value of 0 expresses a relative quantity of search queries for the analysed phrase of less than 1% of its greatest popularity over the given period. All data were accessed and downloaded on 9 November 2020. The Google Trends analysis was based on the methodology framework proposed by Mavragani et al. [27].

### 2.2. Data Acquisition

The division of Poland into 16 voivodeships was used in the spatial analysis (Figure 1). A voivodeship represents the highest-level administrative division of Poland and is a regional administrative unit of Poland [28]. The area, population and population density of each voivodeship was extracted from Statistics Poland (Table 1) [29]. Moreover, in each voivodeship, a capital was selected to investigate potential differences in air quality and SVI values between rural and capitals. The latitude and longitude of each capital were extracted from LatitudeLongitude.org (Table 1) [30].

The data collected by the Chief Inspectorate of Environmental Protection between 2016 and 2019 inclusive were included in the analysis [3]. To analyse seasonal air quality changes, data for daily 24-h (24 h) mean PM_10_ and PM_2.5_ concentrations for each monitoring station between the years 2016 and 2019 were taken into account. Moreover, the number of PM_10_ and PM_2.5_ monitoring stations operating in a given year in a given voivodeship was determined, too (Table 2 and Table 3). In total, for this period, records from 284 monitoring stations reporting PM_10_ concentrations, and 130 monitoring stations reporting PM_2.5_ concentrations were analysed (Figure 1). To analyse spatial relationships, the same data points collected in the same monitoring stations were used. However, due to the availability of straightforward summary data, to minimize the risk of miscalculations, a database that summarises air pollution in Poland for the years 2000–2019 was used [3]. 

Figure 1 and Figure 2 were created by Map Maker [31].

### 2.3. Data Analysis

All data were analysed in Microsoft Office Excel 2016 software developed by Microsoft Corporation.For seasonal analysis, mean monthly GTSVI for all keywords in a given voivodeship were averaged over the whole country (data generated by Google Trends [24]). Using this data four-year monthly GTSVI for all keywords in Poland were calculated (Figure 3a).To conduct spatial analysis, the yearly GTSVI for all keywords in a given voivodeship was computed. Finally, the four-year GTSVI for all keywords in a given voivodeship was determined (Table 1).

With the intention of air quality evaluation, mean 24-h daily concentrations of PM_10_ or PM_2.5_ recorded by a given monitoring station were averaged over all monitoring stations for a given day for a given voivodeship/capital (Figure 3b, data were acquired from the database of the Chief Inspectorate of Environmental Protection [3]). Next, daily measurements were averaged over months. For seasonal analysis, mean monthly PM_10_/PM_2.5_ concentrations data were averaged over the whole country. Using this data four-year monthly PM_10_/PM_2.5_ concentrations for Poland were determined.To conduct spatial analysis, mean monthly data were averaged over years. Finally, the four-year PM_10_/PM_2.5_ concentrations for a given voivodeship/capital were computed (Table 2 and Table 3).

In addition to that, the mean number of monitoring stations per voivodeship, mean area per monitoring station, mean population per monitoring station and mean population density per monitoring station were computed (Table 2 and Table 3).

Corresponding data sets were paired to investigate the spatial and seasonal changes in actual air pollution levels as well as between actual air pollution levels and the popularity of the search queries related to them. Pearson Product-Moment Correlation coefficients (PCC) and p-values were calculated to determine the strength of association between the variables [32,33]. To determine PCC values in the two correlated datasets were always arranged in ascending order. Assessment of the strength of association between the two corresponding datasets was based on the absolute value of PCC. Strength of association was classified as large (PCC > 0.5), medium (0.5 > PCC > 0.3), small (0.3 > PCC > 0.1) or minimal (0.1 > PCC). The threshold of statistical significance was set at *p* < 0.05. 

## 3. Results and Discussion

### 3.1. Spatial Distribution of Air Quality Monitoring Stations

Monitoring stations are more clustered in cities than in rural areas (Figure 1 and Figure 2). It was also found that more polluted regions have a greater number of available air quality monitoring stations as well as that the area per monitoring station is lower in more polluted regions compared to territories that have better air quality (Table 4). The number of active stations per voivodeship also varies, both for PM_10_ and PM_2.5_ monitoring stations (Table 2 and Table 3).

An increase in the mean number of monitoring stations per voivodeship is strongly positively correlated with the incrementation of particulate matter concentrations (PCC > 0.5, *p* < 0.05, Table 4). Conversely, an increase in mean area per monitoring station is strongly or mildly negatively correlated with the increase in particulate matter concentrations.

Neither the increase in mean population per monitoring station nor mean population density per monitoring station is significantly correlated with the rise in PM_10_ or PM_2.5_ concentrations as well as latitude or longitude of the voivodeship’s capital (*p* > 0.05, Table 4).

On the one hand, city-centred distribution of the stations allows for more precise monitoring of air pollution in cities, which have potentially worse air quality due to high traffic and population density. Conversely, such uneven distribution of the measurement locations might have biased calculated mean PM_10_ and PM_2.5_ concentration values. Nevertheless, as in all voivodeships the urban clustering of monitoring stations is observed, all calculations have a similar bias. Therefore, spatial and seasonal analyses based on a comparison of the PM_10_ and PM_2.5_ between the voivodeships can be performed and may be a reasonable source of data.

### 3.2. Spatial Distribution of Air Pollution in Poland

As the latitude of capitals of the voivodeships increases both PM_10_ and PM_2.5_ concentrations recorded in the voivodeship as well as its capital decreases (*p* < 0.05, Table 5). For all four relationships, the strength of association is large. 

With the increase in longitude, minimal changes in PM_10_ concentrations were noticed. Moreover, small increases in PM_2.5_ were observed. However, all relationships are not statistically significant (*p* > 0.05).

The highest mean 2016–2019 PM_10_ concentrations were observed in the Opole (42.6), Cracow (38.5) and Łódź (36.7), while the lowest were observed in the Białystok (20.9), Tricity (22.4) and Szczecin (22.9) capitals (Table 2). Similarly, the highest PM_2.5_ concentrations were observed in the Opole (29.7), Cracow (29.4) and Łódź (24.0) capitals. However, the lowest ones were recorded in the Szczecin (13.5), Tricity (15.8) and Bydgoszcz (15.9) capitals (Table 3).

The highest mean 2016–2019 PM_10_ concentrations were observed in the Śląskie (39.8), Dolnośląskie (35.0) and Łódzkie (34.9) Voivodeships, while the lowest were in the Podlaskie (21.0), Warmińsko-Mazurskie (22.9) and Zachodniopomorskie (23.3) Voivodeships (Table 2). The same trend was recorded for PM_2.5_ concentrations, however, the values were equal to 28.1, 26.8, 25.5 and 15.6, 15.9, 16.2, respectively (Table 3).

Neither any individual capital’s nor any voivodeship’s air quality complies with the WHO air quality guidelines in either annual mean PM_10_ or PM_2.5_ concentrations [4]. Moreover, in the Cracow and Opole capitals, the PM_10_ concentrations were about twice as high as recommended, while the PM_2.5_ concentrations were about three times higher (Table 2 and Table 3). The same situation is observed in the whole of the Małopolskie and Śląskie Voivodeships, where these capitals are located. On the other hand, in other capitals, for example, Białystok, Tricity and Szczecin, as well as their corresponding voivodeships, the mean annual atmospheric PM_10_ concentration was just slightly higher than advised by the guidelines (Table 2). However, the PM_2.5_ concentrations in these locations were still about 50% higher than recommended by the WHO (Table 3).

For the most polluted regions, the data from the Chief Inspectorate of Environmental Protection reports are in line with independently collected data in other studies [34,35]. Moreover, these and other studies [36] show that in these regions submicron PM_1_ concentrations are also elevated and that the smaller the particle, the more toxic it is [36].

In general, the southern regions of the country are more polluted than the northern regions (*p* < 0.05, Table 5) [37]. This gradual improvement in air quality may be caused by increased proximity to the Baltic Sea and the predominantly lowland character of the northern part of the country. These factors allow air to be seamlessly and continuously exchanged [38]. On the other hand, southern regions of Poland are made up of upland, mountainous regions [39] and have a heavily coal-based economy and energy production [40]. In addition, larger cities in the south are located in basins and valleys. Such environmental conditions reduce air movement, cause trapping, and result in the formation of smog and decreased air quality [41]. 

### 3.3. Google Trends Search Volume Index

The greatest mean GTSVI for the years 2016–2019 was observed in the Małopolskie (94.5), Śląskie (75.8) and Dolnośląskie (63.8) Voivodeships. The lowest one was recorded in the Warmińsko-Mazurskie (18.8), Podlaskie (23.3) and Lubuskie (23.8) Voivodeships (Table 1).

### 3.4. Analysis of the Relationships between GTSVI and Air Pollution Levels

Mean voivodeship’s GTSVI is strongly, positively and significantly related to an increase in PM_10_ and PM_2.5_ concentrations in both voivodeships and their capitals (*p* < 0.05, Table 6). Moreover, mean voivodeship’s GTSVI is strongly, negatively and significantly correlated with the increase in latitude. For longitude, no correlation was observed. In addition to that, the rise in mean voivodeship’s GTSVI closely corresponds with the increase in the mean number of monitoring stations per voivodeship (PCC = 0.71, *p* < 0.05) as well as decrementation of mean area per monitoring station (PCC = −0.63, *p* < 0.05).

Neither increase in mean population per monitoring station nor mean population density per monitoring station is significantly correlated with mean voivodeship’s GTSVI (PCC = −0.03, 0.28, respectively; *p* < 0.05).

### 3.5. Spatial Analysis of Poles’ Interest in Air Pollution

Whether people are aware of elevated pollution levels in their local area or not is an important issue. Evidence for this interest can be seen in the high mean Google Trends GTSVI values found in the Cracow or Opole capitals, and the corresponding voivodeships (Table 1). Moreover, a strong linear correlation between the annual level of air pollution in a given area and the frequency of air pollution-related search queries is observed (Table 6) [32]. This means that the higher the mean yearly concentration of PM_10_ and/or PM_2.5_ concentration in a given area, the more frequently people are looking for their local air quality indexes, for example searching online for “air quality” or “air pollution”. It was also found that the Google Trends GTSVI have higher correlations with PM concentrations for capitals’ than for voivodeships’. Of all calculated PCCs, the highest one was observed for the correlation between PM_2.5_ concentration and capitals’ SVI. On the other hand, the weakest correlation was observed for PM_2.5_ concentration and voivodeship’s SVI. The correlations observed for PM_10_ concentration and voivodeship’s GTSVI as well as capitals’ GTSVI are placed in-between the PM_2.5_-related correlations. Thus, it cannot be stated whether changes in PM_2.5_ or PM_10_ concentration are more connected with changes in Poles’ air pollution-related information-seeking behaviour.

### 3.6. Seasonal Analysis of Air Pollution Levels and the Popularity of the Search Queries

The greatest mean monthly GTSVI value was observed for January (29.7), followed by December (26.5) and November (25.6). Conversely, the lowest values were noted for July (3.0), August (3.3) and June (3.8, Figure 4a,b). Mean monthly PM_10_ concentrations [μg·m^−3^] were the highest for January (35.6), February (43.4) and March (47.7), while the lowest were observed for July (17.7), June (19.0) and August (19.1, Figure 4a). The same trend was observed for mean monthly PM_2.5_ concentrations (Figure 4b). The PCC between mean monthly GTSVI and mean monthly PM_10_/PM_2.5_ concentrations were equal to 0.87 (*p* < 0.05) and 0.90 (*p* < 0.05), respectively.

When it comes to seasonal air pollution distribution, the WHO-recommended annual PM_10_ concentrations were met in June, July, and August (Figure 4a,b). On the other hand, in the November–March period, the WHO guidelines were greatly exceeded. Annual PM_2.5_ requirements were not satisfied in any month, but in June, July and August these values were only slightly exceeded. Again, a similar pattern was observed by other researchers [34,35,36]. 

In general, air quality is much worse in winter than in summer which is confirmed by other studies (Figure 4a,b) [34,35]. This happens due to temperature inversion episodes that occur throughout the whole year, but they are more frequent and intensive in the winter season [41,42]. 

Similar to geographical distribution, the greater the seasonal PM_10_ and/or PM_2.5_ concentration, the greater Poles’ interest in air pollution-related search queries. Thus, Poles seem to recognize the importance of checking their local daily air quality index in general, and especially in the winter season when smog is the most harmful [41,42]. Nevertheless, this correlation does not indicate a causal relationship.

During March and April air pollution appears to be underestimated, as evidenced by a relatively lower number of searches for air quality data despite still relatively high pollution levels (Figure 4a,b). On the other hand, November and December data points are situated over the trend line. Therefore, in this period Poles pay more than average attention to air quality relative to the reported levels of air pollution. This may indicate the general public expectation of smog before and during winter. Nonetheless, significant pollution levels are also present during the spring season.

### 3.7. Future Perspectives

Our study as well as others on the same subject show that people are aware of air pollution and execute prevention strategies to protect themselves from potential harms [11,43,44,45]. Interestingly, except for GTSVI people’s response to changing air quality levels can be also monitored via Twitter or Sina Weibo interactions [44,46,47]. Moreover, efforts are being made to inform people about the poor air quality more effectively [48,49]. However, the general public may not link air pollution with its potential to increase the prevalence and/or exacerbation of cardiovascular or respiratory diseases. Therefore, they do not show enough interest in local air quality levels. In addition, society may not recognize the symptoms in an early stage of chronic disease exacerbation. Late admission to the hospital increases the probability of death which results in greater mortality rates [14,15,20,21]. Thus, the general public may not be aware of secondary and tertiary prevention measures, which aim to reduce the impact of a disease that has already occurred [43]. For example, some researchers propose that there is no link between chronic inflammatory skin diseases symptoms and air pollution, which has been shown with the use of Google Trends GTSVI analyses [50,51]. In addition, increased air pollution levels may exacerbate the course of ongoing chronic diseases [52] and affected individuals may not know that. Therefore, starting with chronically ill patients and individuals who live in the most polluted regions worldwide, there is a need for health promotion programs focused on air pollution-related secondary and tertiary prevention strategies, including accurate symptom recognition. Such interventions would result in faster responses to rapid, acute exacerbation of chronic disease symptoms and would probably reduce the cause-specific mortalities related to high air pollution levels, both in the short and long term.

Last but not least, the greatest contributors to the problem of air quality in Poland are the coal-based economy and household heating systems [6]. To reduce the hospital admission and mortality rates potentially caused by air pollution as well as to comply with WHO air quality guidelines, there seems to be a very urgent need for a shift towards more ecological energy production sources.

### 3.8. Limitations

Due to the use of databases, the data were not directly collected by the authors, and thus this study is subject to methodological limitations. The meteorological monitoring stations that measure PM_10_ and PM_2.5_ concentrations were pre-existing and arbitrarily positioned. This may have generated bias in the data, as the concentration of particulate matter is highly dependent on measurement location and locations of the monitoring stations analysed in this study are city-centred. However, as such a pattern is observed in all voivodeships the data are uniformly biased. In addition to that, similar trends in air quality observations were made by other studies which used their measurement stations [34,35,36]. In addition, as mentioned in the Google Trends methodological framework [27] sometimes Google Trends SVIs may differ, even though the keywords used are the same, for example, because the search query was embedded in quotation marks. To reduce this bias, all keywords analysed in this study were always entered without quotation marks, plus signs or any other special characters. Only spaces were used to separate double-word keywords, allowing for the broadest tracking possible. Moreover, Google Trends GTSVI shows the relative, not the absolute number of search queries. However, correlation coefficients can be calculated from either absolute or relative values, thus the calculations in this study remain valid.

Finally, sometimes the data for a given monitoring station were incomplete (below 2% in total). In such a case the average PM_10_/PM_2.5_ concentrations for a given day were calculated from data available from other monitoring stations for this particular voivodeship/capital.

## 4. Conclusions

Neither any capital’s nor any voivodeship’s annual ambient particulate matter concentration complies with the WHO air quality guidelines for either annual mean PM_10_ or PM_2.5_ concentrations. Moreover, in the Cracow and Opole capitals, the PM_10_ concentrations were about twice as high as recommended, while the PM_2.5_ concentrations were about three times higher. It was also found that the south of Poland is more polluted than the north. This gradual improvement in air quality may be caused by increased proximity to the Baltic Sea and the predominantly lowland character of the northern part of the country. 

In addition to that, air quality is much worse in winter than in summer. WHO-advised annual PM_10_ concentrations were satisfied in June, July and August, but the desired PM_2.5_ concentrations were not achieved at any time of the year. In the November-March period, the WHO guidelines were greatly exceeded. However, these conclusions might have been slightly biased due to the uneven distribution of air quality monitoring stations. 

A strong linear correlation between the annual level of air pollution in a given area and the frequency of air pollution-related search queries is observed (PCC ≥ 0.64). It was also found that the GTSVI have higher correlations with PM concentrations for capitals’ than for voivodeships’. Even stronger correlations were found between mean monthly GTSVI values and seasonal PM_10_ and PM_2.5_ quality variations (PCC = 0.87 and 0.90, respectively). Thus, Poles seem to recognize the importance of checking their local daily air quality index in general, and especially in the winter season when smog is the most harmful. As shown by other researches, even better tracking of the societies’ response to elevated air pollution levels as well as a rise in social awareness regarding the issue can be achieved by a smartphone app or information distribution via social media and news reports [44,46,47,48,49].

To conclude, it appears that Poles are aware of the spatial and seasonal changes in air pollution levels, as similar trends were observed for Poles’ interest in air pollution-related keywords. Moreover, the seasonal and regional intensification of their information-seeking behaviour seems to indicate a relevant ad hoc response to the variable threat severity levels. However, the correlations presented in our study do not indicate causal relationships and the need for further research to answer the foremost question of causality is strongly endorsed.

## Figures and Tables

**Figure 1 ijerph-18-11709-f001:**
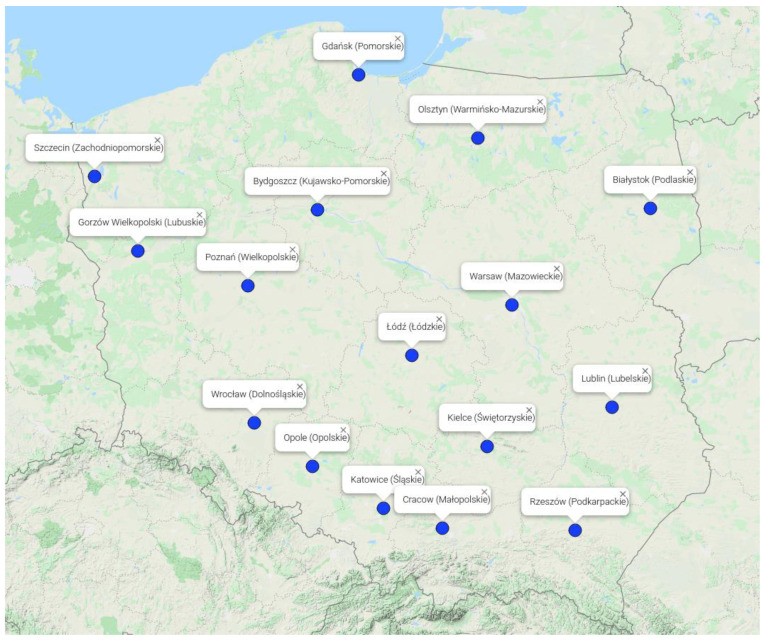
Locations of the capitals of the voivodeships. The name of the voivodeship is provided in brackets.

**Figure 2 ijerph-18-11709-f002:**
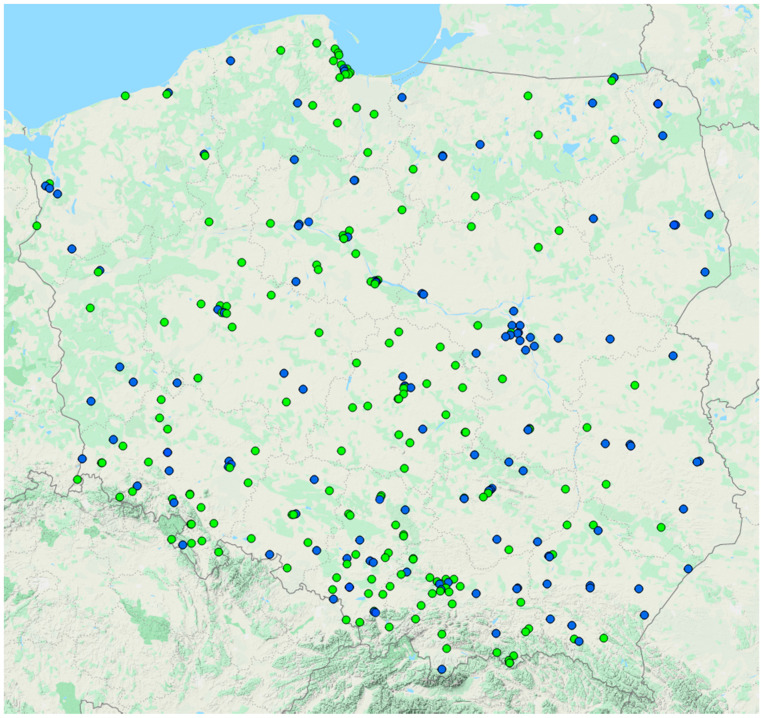
Locations of air quality monitoring stations in Poland. Blue dots represent PM_2.5_ and PM_10_ monitoring stations. Green dots represent the PM_10_ monitoring station.

**Figure 3 ijerph-18-11709-f003:**
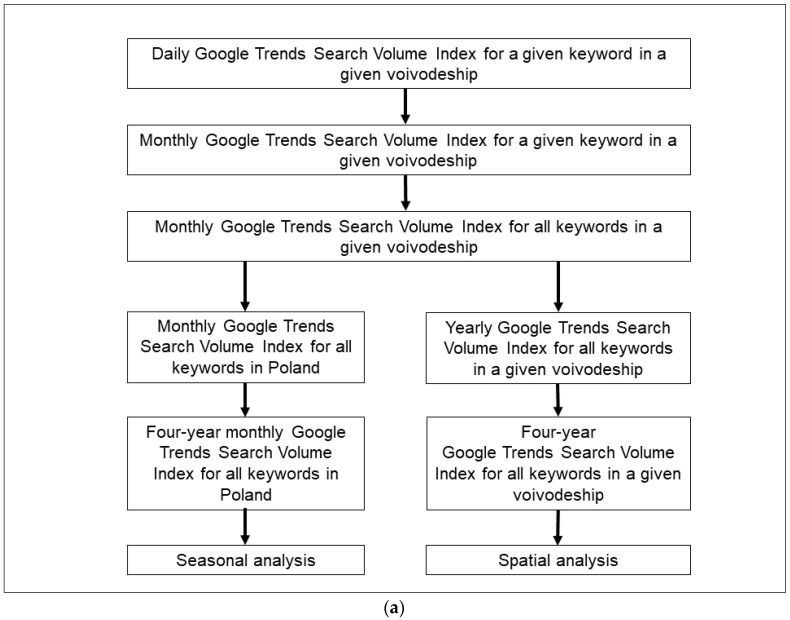
Methodology of GTSVI data synthesis (**a**) and air quality data synthesis (**b**).

**Figure 4 ijerph-18-11709-f004:**
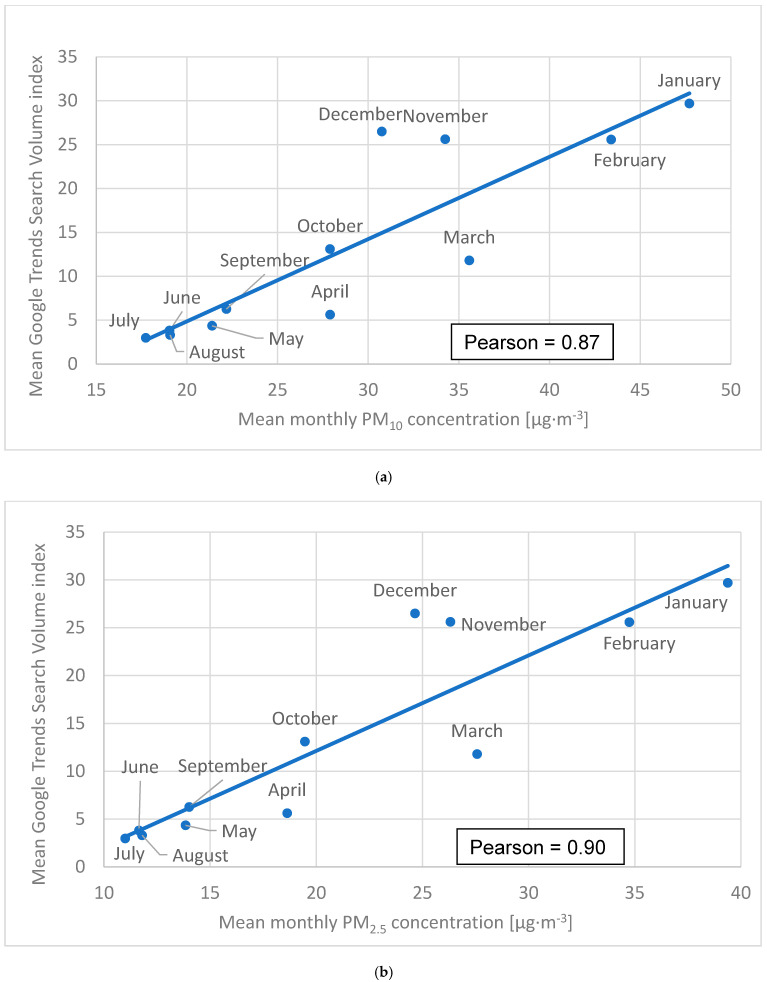
Mean monthly Search Volume Index in relation tomean monthly PM_10_ concentrations (**a**) and mean monthly PM_2.5_ concentrations (**b**).

**Table 1 ijerph-18-11709-t001:** General geographical and demographical characteristics of voivodeships of Poland.

Voivodeship	Area [km^2^]	Population	Population Density [People per km^2^]	Capital of the Voivodeship	Latitude of the Capital (Northern Hemisphere)	Longitude of the Capital (Eastern Hemisphere)	Mean Voivodeship’s GTSVI for Years 2016–2019
Warmińsko-Mazurskie	24,173	1,422,737	59	Olsztyn	53.78	20.49	18.75
Podlaskie	20,187	1,178,353	58	Białystok	53.13	23.16	23.25
Lubuskie	13,988	1,011,592	72	Gorzów Wielkopolski	52.74	15.23	23.75
Zachodnio pomorskie	22,905	1,696,193	74	Szczecin	53.43	14.55	24.25
Pomorskie	18,323	2,343,928	128	Gdańsk	54.35	18.65	24.25
Lubelskie	25,123	2,108,270	84	Lublin	51.25	22.57	28.00
Kujawsko-Pomorskie	17,971	2,072,373	115	Bydgoszcz	53.12	18.01	28.25
Podkarpackie	17,846	2,127,164	119	Rzeszów	50.04	22.00	31.00
Wielkopolskie	29,826	3,498,733	117	Poznań	52.41	16.93	33.50
Łódzkie	18,219	2,454,779	135	Łódź	51.75	19.47	34.50
Opolskie	9412	982,626	104	Opole	50.67	17.93	42.25
Świętorzyskie	11,710	1,233,961	105	Kielce	50.87	20.63	45.50
Mazowieckie	35,559	5,423,168	153	Warsaw	52.23	21.01	58.50
Dolnośląskie	19,947	2,900,163	145	Wrocław	51.10	17.03	63.75
Śląskie	12,333	4,517,635	366	Katowice	50.26	19.03	75.75
Małopolskie	15,183	3,410,901	225	Cracow	50.06	19.94	94.50
Poland	19,544	2,398,911	129	-	-	-	-

**Table 2 ijerph-18-11709-t002:** General characteristics of PM_10_ monitoring stations operating between years 2016–2019 in Poland.

Voivodeship	Data for PM_10_ Monitoring Stations (Means for Years 2016–2019)										
	Number of Monitoring Stations Operating in a Given Year				Total Number of Operating Monitoring Stations Operating at Any Time during the Analysed Period	Mean Number of Monitoring Stations per Voivodeship	Mean Area per Monitoring Station [km^2^ of Land Area]	Mean Population per Monitoring Station	Mean Population Density per Monitoring Station	Mean PM_10_ Concentration for a Voivodeship	Mean PM_10_ Concentration for a Capital City
	2016	2017	2018	2019							
Warmińsko-Mazurskie	9	9	9	10	12	9.3	2014	118,561	5	22.9	23.3
Podlaskie	6	6	6	6	8	6.0	2523	147,294	7	21.0	20.9
Lubuskie	6	6	7	7	7	6.5	1998	144,513	10	26.2	24.5
Zachodniopomorskie	9	9	10	10	11	9.5	2082	154,199	7	23.3	22.4
Pomorskie	21	19	18	16	21	18.5	873	111,616	6	23.9	22.9
Lubelskie	8	8	9	10	11	8.8	2284	191,661	8	27.3	25.5
Kujawsko-Pomorskie	17	19	18	18	23	18.0	781	90,103	5	28.0	33.4
Podkarpackie	11	13	14	16	17	13.5	1050	125,127	7	28.7	28.4
Wielkopolskie	15	15	16	17	17	15.8	1754	205,808	7	30.4	28.4
Łódzkie	23	25	24	24	26	24.0	701	94,415	5	34.9	36.7
Opolskie	8	8	8	9	10	8.3	941	98,263	10	31.3	30.4
Świętorzyskie	8	10	11	10	13	9.8	901	94,920	8	28.4	31.6
Mazowieckie	20	21	19	22	24	20.5	1482	225,965	6	29.0	32.8
Dolnośląskie	23	23	25	25	32	24.0	623	90,630	5	28.5	28.5
Śląskie	24	24	25	24	27	24.3	457	167,320	14	39.8	42.6
Małopolskie	25	25	27	26	37	25.8	410	92,187	6	35.0	38.5
Mean for a voivodeship	15	15	15	16	19	15	1305	134,536	7	28.7	29.4

**Table 3 ijerph-18-11709-t003:** General characteristics of PM_2.5_ monitoring stations operating between years 2016–2019 in Poland.

Voivodeship	Data for PM_2.5_ Monitoring Stations (Means for Years 2016–2019)										
	Number of Monitoring Stations Operating in a Given Year				Total Number of Operating Monitoring Stations Operating at Any Time during the Analysed Period	Mean Number of Monitoring Stations per Voivodeship	Mean Area per Monitoring Station [km^2^ of Land Area]	Mean Population per Monitoring Station	Mean Population Density per Monitoring Station	Mean PM_2.5_ Concentration for a Voivodeship	Mean PM_2.5_ Concentration for a Capital City
	2016	2017	2018	2019							
Warmińsko-Mazurskie	4	4	4	5	6	4.3	4029	237,123	10	15.6	16.7
Podlaskie	6	6	6	6	8	6.0	2523	147,294	7	21.5	17.5
Lubuskie	4	4	5	5	5	4.5	2798	202,318	14	18.0	17.4
Zachodniopomorskie	5	5	5	6	6	5.3	3818	282,699	12	16.2	13.5
Pomorskie	4	4	4	4	4	4.0	4581	585,982	32	15.9	15.8
Lubelskie	5	5	5	6	7	5.3	3589	301,181	12	21.5	18.7
Kujawsko-Pomorskie	8	10	8	9	11	8.8	1634	188,398	10	17.8	15.9
Podkarpackie	8	9	9	11	12	9.3	1487	177,264	10	22.5	21.5
Wielkopolskie	4	4	4	4	4	4.0	7457	874,683	29	25.2	21.6
Łódzkie	5	5	5	5	5	5.0	3644	490,956	27	25.5	24.0
Opolskie	3	3	3	4	4	3.3	2353	245,657	26	20.1	21.3
Świętorzyskie	6	6	7	6	9	6.3	1301	137,107	12	21.2	21.8
Mazowieckie	14	15	14	17	18	15.0	1976	301,287	8	20.9	19.6
Dolnośląskie	8	8	8	11	11	8.8	1813	263,651	13	19.5	20.1
Śląskie	9	9	10	10	11	9.5	1121	410,694	33	28.1	29.7
Małopolskie	9	9	9	9	9	9.0	1687	378,989	25	26.8	29.4
Mean for a voivodeship	6	7	7	7	8	6.8	2863	326,580	18	21.0	20.3

**Table 4 ijerph-18-11709-t004:** Pearson product-moment correlation coefficients for correlations between geographic coordinates of capital cities of voivodeships and corresponding particulate matter concentration changes.

Analysed Factor	Latitude	*p*-Value	Longitude	*p*-Value
PM_10_ concentration in a voivodeship	−0.73	**<0.01**	−0.04	0.90
PM_10_ concentration in a capital city	−0.65	**<0.01**	0.05	0.85
PM_2.5_ concentration in a voivodeship	−0.71	**<0.01**	0.28	0.29
PM_2.5_ concentration in a capital city	−0.80	**<0.01**	0.21	0.43

Data in bold indicate statistically significant associations.

**Table 5 ijerph-18-11709-t005:** Pearson product-moment correlation coefficients for correlations between aerial distribution of monitoring stations and corresponding variables.

Analysed Factors	Mean Number of Monitoring Stations per Voivodeship	*p*-Value	Mean Area per Monitoring Station [km^2^]	*p*-Value	Mean Population per Monitoring Station	*p*-Value	Mean Population Density per Monitoring Station	*p*-Value
PM_10_ concentration in a voivodeship	0.68	**<0.01**	−0.74	**<0.01**	0.12	0.65	0.38	0.15
PM_10_ concentration in a capital city	0.73	**<0.01**	−0.77	**<0.01**	−0.17	0.53	0.28	0.29
PM_2.5_ concentration in a voivodeship	0.51	**0.04**	−0.39	0.14	0.15	0.57	0.36	0.17
PM_2.5_ concentration in a capital city	0.60	**0.02**	−0.60	**0.01**	−0.08	0.76	0.43	0.09
Latitude	−0.31	0.24	0.48	0.06	0.07	0.80	−0.49	0.06
Longitude	−0.06	0.82	0.23	0.40	0.28	0.30	0.05	0.85

Data in bold indicate statistically significant associations.

**Table 6 ijerph-18-11709-t006:** Pearson product-moment correlation coefficients for correlations between mean voivodeship’s GTSVI for years 2016–2019 and corresponding variables.

Analysed Factors	Mean Voivodeship’s GTSVI for Years 2016–2019	*p*-Value
PM_10_ concentration in a voivodeship	0.73	**<0.01**
PM_10_ concentration in a capital city	0.76	**<0.01**
PM_2.5_ concentration in a voivodeship	0.64	**0.01**
PM_2.5_ concentration in a capital city	0.81	**<0.01**
Latitude	−0.68	**<0.01**
Longitude	0.04	0.87
Mean number of monitoring stations per voivodeship	0.71	**<0.01**
Mean area per monitoring station [km^2^ of land area]	−0.63	**0.01**
Mean population per monitorng station	−0.03	0.90
Mean population density per monitoring station	0.28	0.29

Data in bold indicate statistically significant associations.

## Data Availability

The data presented in this study are available on request from the corresponding author.

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
