# Peer review of "Changes in Air Pollution-Related Behaviour Measured by Google Trends Search Volume Index in Response to Reported Air Quality in Poland"

_ijerph, 2021, doi:10.3390/ijerph182111709_

Round 1
Reviewer 1 Report
This manuscript is a novel research based on the google trends search volume. I suggest that the paper can be published in this journal after major revision.

Author Response
###Dear Reviewer, thank you for your comments. We appreciate your time and effort dedicated to improve our research paper. You can find direct response to your questions and suggestions below:
This manuscript is a novel research based on the google trends search volume. I suggest that the paper can be published in this journal after major revision.
Keywords: Add a search volume index
###Added.
L33, PM2.5 and PM10 in the full text are changed to PM2.5 and PM10.
###Changed in the text.
L36, The unit of particle concentration is changed to the multiplication form, μg·m-3 .
###Changed in the text.
L39-40, 46, The discussion need to be supported by relevant citations or official government websites.
###Citations were added.
L61, Google Trends Search Volume Index is abbreviated to GTSVI. When it appears again below, GTSVI can be directly used. The Google Trends Search Volume Index in the tables or figures can all be written as GTSVI.
###Changed as requested
L65-l71, L105-111, L118, L130, can be combined into one paragraph.
###We left the aim of the study as the last paragraphs of the introduction. Also, in our opinion, the aim of the study does not fit well to the “materials and methods” section.
The other paragraphs were merged and placed in the “materials and methods” section, as requested.
L76, search engine optimization (SEO).
###Changed as requested
L96, Search volume index (SVI).
###Changed as requested
L114, L134, Comments on a figure can be placed directly after the figure name or annotated within the figure.
###Changed as requested
L116, km2
###Changed as requested
L136 By merging Tables 2 and 3, the contents of the first row and the first column of the two tables are almost identical.
###We left the Tables 2 and 3 unchanged, as if they were merged it would be hard to put them on one page – that would be inconvenient for the reader.
L139-140 Delete this section.
###Changed as requested
L163, Merge figures 3 and 4, and merge figures of the same type to reduce the number of figures. Mark them with a or b to distinguish small figures.
###Changed as requested
L178 Results and discussion should be put together. In a paragraph, describe the results first and then write the discussion.
###Changed as requested
L249, Merge Figure 5 and Figure 6. The coordinate name of the figure needs to be capitalized. Mean monthly
###Changed as requested
L254-260, This part is exactly the same as L65-71, we suggest deleting it.
###Changed as requested
L270, There's something wrong with the font.
###Corrected
L336-363, This is the part of the introduction.
###With minor changes this part was added to introduction
L364-420 To abbreviate these three parts into one conclusion. The future prospects and limitations of the experiment should not be described in a large amount of space size in conclusion section. But some opinions can be interspersed into the results and discussion
###We left the future perspectives and limitations unchanged, as the 2nd reviewer highly endorsed a need for these. Conclusions were developed.
Reviewer 2 Report
I found the paper very interesting, and the authors analyzed their ideas and methodology up to a great degree. It was also very nice to see a “Future perspective” and “Limitations” sections. However, the paper should be considered for publication after considering the following notes first:
- Rewrite the abstract as it is somehow confusing in some parts. It’s also a good idea to state the contribution of the paper clearer.
- Improve the literature review: The papers mentioned are important in that field, but you should provide more papers to better support your study.
- At the data analysis section, you should also cite the websites you took the data from.
- Increase the concluding remarks to incorporate more details from the paper.
- Improve the language used.
Author Response
###Dear Reviewer, thank you for your comments. We appreciate your time and effort dedicated to improve our research paper. You can find direct response to your questions and suggestions below:
I found the paper very interesting, and the authors analyzed their ideas and methodology up to a great degree. It was also very nice to see a “Future perspective” and “Limitations” sections. However, the paper should be considered for publication after considering the following notes first:
- Rewrite the abstract as it is somehow confusing in some parts. It’s also a good idea to state the contribution of the paper clearer.
###Abstract was rewritten. The contributions are more detailed now.
- Improve the literature review: The papers mentioned are important in that field, but you should provide more papers to better support your study.
###We added 6 new references.
- At the data analysis section, you should also cite the websites you took the data from.
###We added citations at relevant places.
- Increase the concluding remarks to incorporate more details from the paper.
###The conclusion section was greatly expanded.
- Improve the language used.
###Text was proofread by a certified medical proofreader.
Reviewer 3 Report
The authors relate air quality is related to increased daily mortality rates by analysing spatial and seasonal changes in air pollution-related information-seeking behaviour in response to nationally reported air quality in Poland.
The article is very relevant and pertinent. I would like the authors to answer some questions and make some small changes to improve the quality of the article. Here are my suggestions:
1. Justifying why the authors take data on PM10 and PM2.5 concentrations measured in the whole of Poland between 2016 and 2019.
2. Justify why the Pearson product-moment correlation coefficients statistic has been used to measure the strength of spatial and seasonal relationships between reported air pollution levels and the popularity of search queries.
3. Justify why the results showed higher concentrations of PM10 and PM2.5 in the south during the winter season? Can this be extrapolated to other cities?
4. In the conclusions, state how Poles can be made aware of air quality levels.
5. Justify why the conclusions might be slightly biased due to the uneven distribution of air quality monitoring stations? What level of significance and percentage error support the results.
6. Enrich the state of the art with articles on the following topics.
Gurajala, S., Dhaniyala, S., & Matthews, J. N. (2019). Understanding Public Response to Air Quality Using Tweet Analysis. Social Media and Society, 5(3). https://doi.org/10.1177/2056305119867656. https://doi.org/10.1177/2056305119867656.
Delmas, M. A., & Kohli, A. (2020). Can Apps Make Air Pollution Visible? Learning About Health Impacts Through Engagement with Air Quality Information. Journal of Business Ethics, 161(2), 279-302. https://doi.org/10.1007/s10551-019-04215-7
Domínguez-amarillo, S., Fernández-agüera, J., Cesteros-garcía, S., & González-lezcano, R. A. (2020). Bad air can also kill: Residential indoor air quality and pollutant exposure risk during the covid-19 crisis. International Journal of Environmental Research and Public Health, 17(19), 1-34. https://doi.org/10.3390/ijerph17197183
D'Antoni, D., Auyeung, V., Walton, H., Fuller, G. W., Grieve, A., & Weinman, J. (2019). The effect of evidence and theory-based health advice accompanying smartphone air quality alerts on adherence to preventative recommendations during poor air quality days: A randomised controlled trial. Environment International, 124, 216-235. https://doi.org/10.1016/j.envint.2019.01.002
Hormigos-Jimenez, S., Padilla-Marcos, M. Á., Meiss, A., Gonzalez-Lezcano, R. A., & Feijó-Muñoz, J. (2018). Computational fluid dynamics evaluation of the furniture arrangement for ventilation efficiency. Building Services Engineering Research and Technology, 39(5), 557-571. https://doi.org/10.1177/0143624418759783.
7. The conclusions are a bit poor. Indicate by means of milestones all the objectives achieved and future lines of research as well as future recommendations.
Author Response
###Dear Reviewer, thank you for your comments. We appreciate your time and effort dedicated to improve our research paper. You can find direct response to your questions and suggestions below:
The authors relate air quality is related to increased daily mortality rates by analysing spatial and seasonal changes in air pollution-related information-seeking behaviour in response to nationally reported air quality in Poland.
The article is very relevant and pertinent. I would like the authors to answer some questions and make some small changes to improve the quality of the article. Here are my suggestions:
1. Justifying why the authors take data on PM10 and PM2.5 concentrations measured in the whole of Poland between 2016 and 2019.
###We covered that in our manuscript: “This period was selected as at the beginning of 2016 Google introduced an improvement to their Google Trends data collection system [24]. 2020 was not included as the ongoing COVID-19 pandemic [26] may have affected people’s behaviour which would result in biased data.”
2. Justify why the Pearson product-moment correlation coefficients statistic has been used to measure the strength of spatial and seasonal relationships between reported air pollution levels and the popularity of search queries.
### The Pearson Product Moment coefficient is a good way “to measure the statistical relationship, or association, between two continuous variables. It is known as the best method of measuring the association between variables of interest because it is based on the method of covariance. It gives information about the magnitude of the association, or correlation, as well as the direction of the relationship.”. Therefore, it was used in our study
Source: https://www.statisticssolutions.com/free-resources/directory-of-statistical-analyses/pearsons-correlation-coefficient/
Justify why the results showed higher concentrations of PM10 and PM2.5 in the south during the winter season? Can this be extrapolated to other cities?
###We covered that in our paper: “In general, the southern regions of the country are more polluted than the northern regions (p<0.05, Table 5) [37]. This gradual improvement in air quality may be caused by increased proximity to the Baltic Sea and the predominantly lowland character of the northern part of the country. These factors allow air to be seamlessly and continuously exchanged [38]. On the other hand, southern regions of Poland are made up of upland, mountainous regions [39] and have a heavily coal-based economy and energy production [40]. Also, larger cities in the south are located in basins and valleys. Such environmental conditions reduce air movement, cause trapping, and result in the formation of smog and decreased air quality [41].
”
We think that it can be extrapolated to other cities.
4. In the conclusions, state how Poles can be made aware of air quality levels.
###we added: “As shown by other researches, an even better response to elevated air pollution levels can be probably achieved by a smartphone app, information distribution via social media or news reports”
5. Justify why the conclusions might be slightly biased due to the uneven distribution of air quality monitoring stations? What level of significance and percentage error support the results.
###We covered that in our paper: “. The meteorological monitoring stations that measure PM10 and PM2.5 concentrations were pre-existing and arbitrarily positioned. This may have generated bias in the data, as the concentration of particulate matter is highly dependent on measurement location and locations of the monitoring stations analysed in this study are city-centred. However, as such a pattern is observed in all voivodeships the data are uniformly biased.”
Unfortunately, it is hard to assess the bias exactly. The good is that the bias is more less uniform, which means that if we compare the data relatively to each other (for example by calculating Pearson Product Moment Coefficient) than the bias is very small or there can be no bias at all.
Enrich the state of the art with articles on the following topics.
### We cited some of the provided articles. Thank you for your suggestions.
- The conclusions are a bit poor. Indicate by means of milestones all the objectives achieved and future lines of research as well as future recommendations.
### Conclusions were greatly extended. Future recommendations and research perspectives are covered in the “future perspectives” section.
Round 2
Reviewer 3 Report
The authors have answered the questions posed with solvency and have made corrections to the article that have notably improved its quality.